# ENGRAM: EFFECTIVE, LIGHTWEIGHT MEMORY ORCHESTRATION FOR CONVERSATIONAL AGENTS

## ABSTRACT

Large language models (LLMs) deployed in user-facing applications require long-horizon consistency: the capacity to remember prior interactions, respect user preferences, and ground reasoning in past events. However, contemporary memory systems often adopt complex architectures such as knowledge graphs, multi-stage retrieval, and operating-system–style schedulers, which introduce engineering complexity and reproducibility challenges. We present ENGRAM, a lightweight state-of-the-art memory system that organizes conversation into three canonical memory types—episodic, semantic, and procedural—through a single router and retriever. Each user turn is converted into typed memory records with normalized schemas and embeddings and persisted in a database. At query time, the system retrieves top-k dense neighbors per type, merges results with simple set operations, and provides relevant evidence as context to the model. ENGRAM attains state-of-the-art results on the LoCoMo benchmark, a realistic multi-session conversational question-answering (QA) suite for long-horizon memory, and exceeds the full-context baseline by 15 absolute points on Long-MemEval, an extended-horizon conversational benchmark, while using only $\approx 1\%$ of the tokens. Our results suggest that careful memory typing and straightforward dense retrieval enable effective long-term memory management in language models, challenging the trend toward architectural complexity in this domain.

## 1 INTRODUCTION

LLMs are now embedded in personal assistants, tutoring systems, productivity tools and many other user-facing applications. These deployments demand long-horizon consistency, meaning remembering prior interactions, respecting user preferences, and grounding reasoning in past events (Baddeley & Hitch, 1974). However, unlike humans, LLMs reset once input falls outside the context window. This leads to brittle behaviors such as forgetting, contradictions, or reliance on pre-training (Liu et al., 2023). Prior efforts extended the Transformer architecture (Vaswani et al., 2017) to increase effective context, but they do not obviate the need for external memory (Dai et al., 2019; Beltagy et al., 2020).

Contemporary memory systems have converged on increasingly elaborate architectures, often involving knowledge graphs, multi-stage retrieval pipelines, or operating-system style schedulers. These designs introduce engineering complexity and many degrees of freedom, making reproducibility and analysis difficult. In contrast, we argue for a different point on the design spectrum: a compact memory layer that is intentionally simple yet sufficient to deliver state-of-the-art accuracy and reliable performance.

We introduce ENGRAM, a lightweight memory system that separates **three memory types** — *episodic*, *semantic*, **and** *procedural* — and combines them through a single router and a single retriever. Each user message is converted into typed memory records with normalized JSON schemas and embeddings. Records are persisted in a local SQLite store, and at query time the system retrieves top-k neighbors from each memory type, merges results, and provides the evidence set as context to the answering prompt (Guu et al., 2020).

Our central claim is that careful memory typing, minimal routing, and straightforward retrieval suffice to achieve **state-of-the-art performance** on LoCoMo, and to **surpass full-context baselines** on LongMemEval (Maharana et al., 2024; Wu et al., 2025). We demonstrate consistent gains across

single-hop, multi-hop, open-domain, and temporal categories. Beyond headline metrics, the simplicity of ENGRAM makes it an attractive foundation for principled experimentation: each component is small, interpretable, and easy to swap out. Finally, we provide a complete implementation and evaluation harness to support rigorous comparison and foster adoption by the community.

## 2 RELATED WORK

Existing work on long-term memory for language models spans non-parametric retrieval methods (Khandelwal et al., 2020), graph-based structures, and system-level abstractions. Non-parametric approaches augment a model with an external store accessed through dense or lexical retrieval (Lewis et al., 2020; Guu et al., 2020). Nearest-neighbor LMs and large-scale retrieval-pretraining extend this line (Borgeaud et al., 2022). They are attractive for freshness and editability, yet often depend on retriever calibration and heuristic chunking. Graph-based methods organize memories as nodes and relations to support structured traversal and multi-hop reasoning (Anokhin et al., 2024; Li et al., 2024). These designs capture compositional structure but introduce orchestration complexity and latency overhead at inference time. System-level approaches treat memory as a schedulable resource, adding lifecycle management and governance primitives (Packer et al., 2023; Li et al., 2025).

Our work is closest to recent memory modules aimed at practical deployment. In contrast to multi-layer schedulers and heavy graph construction, ENGRAM retains the benefits of typed memory and semantic retrieval while **minimizing moving parts**, yielding a system that is straightforward to operate in both research and production at scale.

## 3 THE ENGRAM ARCHITECTURE

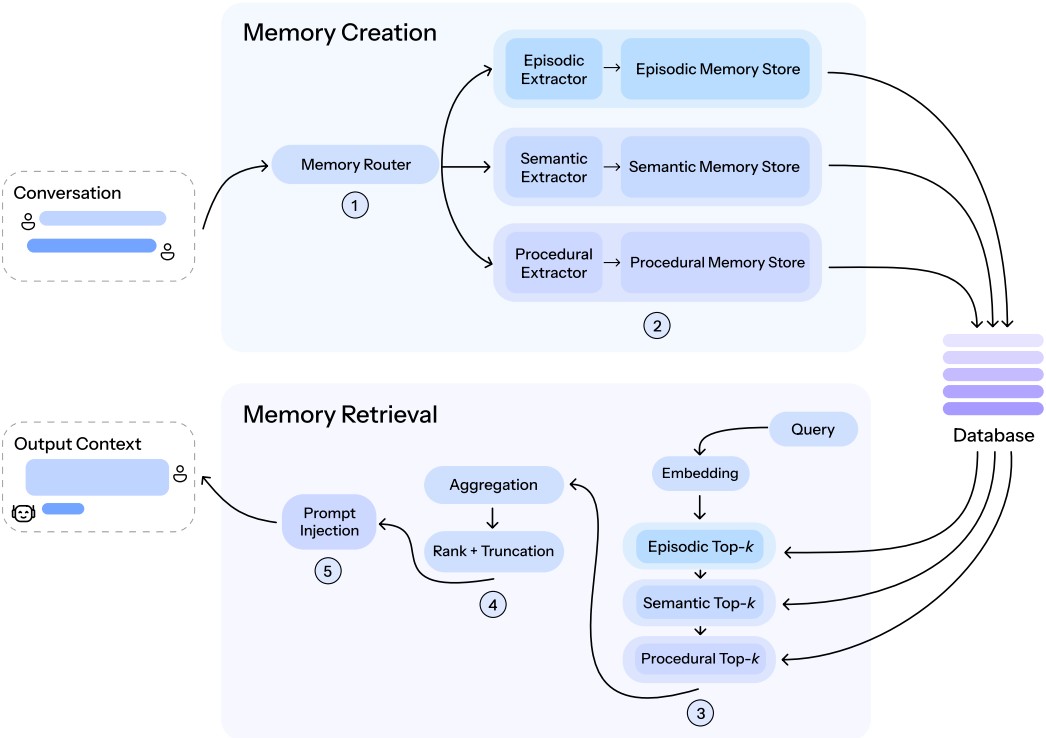

Figure 1: **System overview of ENGRAM.** Turns are routed into typed stores, embedded, persisted, and later retrieved with semantic search before being passed as context to an answering model. The diagram highlights both the memory creation stage (routing and extraction) and the retrieval stage (top-$k$ selection, aggregation, and prompt injection). Numbers (1)–(5) mark the main components and are referenced below.

A router (1) determines which memory buckets apply to an incoming utterance. For each selected bucket, a lightweight extractor converts the utterance into a normalized record and requests an embedding (2). Records are persisted in SQLite together with their embeddings. At query time, the system retrieves the top-$k$ items by cosine similarity from all buckets (3), merges and deduplicates the results (4), and supplies the selected snippets as context to the answering model (5). The stages (1)–(5) together with the overall architecture are shown in Figure 1. We formalize each stage in the subsections that follow, and provide an end-to-end QA walkthrough in Appendix C.

### 3.1 PROBLEM SETUP

We model a dialogue as a sequence of turns

$$\mathcal{C} = \{x_t\}_{t=1}^T, \quad x_t = (s_t, u_t, \tau_t)$$

where $s_t \in \{A, B\}$ denotes the speaker identity, $u_t \in \mathcal{U}$ is the turn text drawn from the space of natural language, and $\tau_t \in \mathbb{R}_+$ is a temporal marker. ENGRAM learns a mapping $f : \mathcal{C} \mapsto \mathcal{M}$ that transforms a dialogue $\mathcal{C}$ into a durable memory state $\mathcal{M}$ capable of supporting answers to questions $q \in \mathcal{U}$ posed after the conversation has unfolded.

### 3.2 ROUTING AND STORAGE

Every turn must be mapped into one or more memory types. ENGRAM employs a router (1) that decides which of the three stores a turn $u_t$ should update, represented as a compact three-bit mask $b_t$

$$r(u_t) \in \{0,1\}^3 \quad \Rightarrow \quad b_t = \left(b_t^{\text{epi}}, b_t^{\text{sem}}, b_t^{\text{pro}}\right)$$

When the router outputs a one for a given type $k \in \{\text{epi}, \text{sem}, \text{pro}\}$, the turn is written to that store. For each selected type $k$, the system creates a structured record from the turn and pairs it with an embedding vector $e \in \mathbb{R}^d$ produced by the encoder $g : \mathcal{U} \to \mathbb{R}^d$. Each memory record therefore has two parts: an interpretable set of fields (e.g., text and timestamp) and a dense vector representation suitable for similarity search. Because the router produces a compact three-bit mask, the routing process remains both interpretable and easy to ablate.

### 3.3 TYPED MEMORY STORES

Once turns are routed and stored, ENGRAM organizes them into three typed stores (2): episodic $m^{\text{epi}}$, semantic $m^{\text{sem}}$, and procedural $m^{\text{pro}}$. Episodic memory encodes events that unfold in time, semantic memory preserves stable facts or preferences, and procedural memory retains instructions or workflows (Tulving, 1972; Cohen & Squire, 1980). Formally, we represent records in each memory store as

$$m^{\text{epi}} = (t, \sigma, \delta, e), \quad m^{\text{sem}} = (f, \delta, e), \quad m^{\text{pro}} = (t, c, \delta, e)$$

where $t$ is a concise title, $\sigma$ a short summary, $\delta$ a temporal anchor, and $e \in \mathbb{R}^d$ an embedding vector from $g$. For semantic records, $f$ is a fact string, and for procedural records, $c$ is normalized content that may correspond to multi-step instructions. These typed records populate the memory state for a user

$$\mathcal{M} = \left(\mathcal{M}_{\text{epi}}, \mathcal{M}_{\text{sem}}, \mathcal{M}_{\text{pro}}\right)$$

where each $\mathcal{M}_k$ is a finite sequence of records of the corresponding schema—episodic $(t, \sigma, \delta, e)$, semantic $(f, \delta, e)$, and procedural $(t, c, \delta, e)$. Here, the typed separation constrains extraction, reduces competition at retrieval, and exposes structure that can be directly inspected by researchers or downstream models.

## 3.4 DENSE RETRIEVAL

Given the memory state $\mathcal{M}$, the system must retrieve relevant records at query time. A query $q$ is embedded as $e_q = g(q)$. For each store $k \in \{\mathrm{epi}, \mathrm{sem}, \mathrm{pro}\}$ and record $m \in \mathcal{M}_k$, ENGRAM computes cosine similarity and selects the top-$k$ items within that store (3) (Karpukhin et al., 2020). These selected memories represent the most relevant items per type (e.g., events from episodic, facts from semantic, or instructions from procedural memory). The retrieved records are then merged and deduplicated across stores. Finally, the combined results are truncated (4) to a fixed budget of $K=25$, an intentional choice motivated by our ablation analysis (see Appendix A.2).

$$R_k(q) = \mathrm{TopK}\big\{\mathrm{score}(q, m) \mid m \in \mathcal{M}_k\big\}$$

$$\tilde{R}(q) = \mathrm{Truncate}_K\Big(\mathrm{Dedup}\bigcup_k R_k(q)\Big)$$

$R_k(q)$ denotes the top-$k$ records retrieved from store $k$, and $\tilde{R}(q)$ is the final set after merging, deduplication, and truncation. The result $\tilde{R}(q)$ is the set of memories passed forward to the answer generation stage.

## 3.5 ANSWER GENERATION

At this stage, the retrieved memories are organized into speaker-specific banks to handle multi-speaker settings. Given a query $q$ about a dialogue between speakers $A$ and $B$, the system produces $\tilde{R}(q, A)$ and $\tilde{R}(q, B)$. Each record $m$ is serialized as

$$\ell(m) = \delta(m) : \mathrm{text}(m)$$

which produces a compact representation $\ell(m)$ that aligns the temporal anchor $\delta(m)$ with the corresponding textual content $\mathrm{text}(m)$. To construct the final input for the model, we define $\mathrm{Template}$ as a fixed, non-learned formatting function that deterministically combines the query and serialized records into a natural-language prompt. The final prompt is then assembled (5) to include memory records from both speakers:

$$P(q) = \mathrm{Template}\Big(q, \ \{\ell(m)\}_{m \in \tilde{R}(q,A)}, \ \{\ell(m)\}_{m \in \tilde{R}(q,B)}\Big)$$

This prompt is passed to the language model to produce the answer $\hat{a} = \mathrm{LLM}(P(q))$. Separating speaker-specific banks ensures that evidence remains properly attributed, avoids conflating voices, and allows disambiguation when multiple interlocutors are present. This completes the end-to-end pipeline described in Section 3.

## 4 EVALUATION

We evaluate ENGRAM on two complementary long-horizon conversational benchmarks. LoCoMo compresses realistic two-speaker dialogues into long, multi-session conversations that probe diverse reasoning categories. LongMemEval instead embeds QA tasks in extended user–assistant histories, stressing durability, updates, and abstention. Our evaluation covers dataset-specific preprocessing, answer-quality and retrieval metrics, a principled baseline suite, and latency analysis. Numerical results appear in the next section.

### 4.1 BENCHMARKS

**LoCoMo** LoCoMo comprises long-term multi-session dialogues constructed via a human–machine pipeline grounded in personas and event graphs, followed by human edits for long-range consistency (Maharana et al., 2024). The released benchmark contains 10 dialogues, each averaging $\approx 600$ turns and $\approx 16\mathrm{K}$ tokens across up to 32 sessions. The QA split labels questions into five

categories: single-hop, multi-hop, temporal, commonsense/world knowledge, and adversarial/unanswerable. Following common practice, we exclude adversarial/unanswerable items when reporting QA metrics and provide category-wise breakdowns for the remaining four types.

**LongMemEval** LongMemEval targets interactive memory in user–assistant settings and evaluates five core abilities (Wu et al., 2025): information extraction, multi-session reasoning, temporal reasoning, knowledge updates, and abstention (i.e. declining to answer when evidence is insufficient). The benchmark provides 500 curated questions embedded in length-configurable chat histories. We evaluate on LongMemEval$_S$ ($\approx$ 115K tokens per problem) and report QA metrics.

## 4.2 METRICS

We report a suite of metrics that capture semantic correctness, lexical fidelity, retrieval quality, and efficiency.

**F1 and B1 (lexical fidelity)**. We report token-level F1 and BLEU-1/2 or B1-1/2 with smoothing as conventional indicators of surface-level agreement (Papineni et al., 2002). Prior to scoring, predictions and references undergo deterministic normalization (Unicode NFKC, lower-casing, removal of articles and punctuation, whitespace compaction, light numeric canonicalization and resolution of relative temporal forms to ISO-8601). These metrics quantify wording overlap and enable comparability with prior work, but they are insensitive to factual inversions (e.g., "Fiona was born in March" vs. "Fiona is born in February" yields high overlap despite a critical error in semantic meaning). Consequently, F1/B1 are treated as complementary diagnostics rather than primary measures of correctness.

**LLM-as-a-Judge (semantic correctness)**. To capture factual accuracy beyond lexical overlap, we employ an independent LLM that, given the question, gold answer(s), and prediction, renders a binary semantic-correctness decision based on factual fidelity, relevance, completeness, and contextual appropriateness. We use GPT-4O-MINI as the judging model (OpenAI, 2023; Zheng et al., 2023). Because judge inferences are stochastic, each method is evaluated three times over the full test set and we report the mean $\pm$ one standard deviation. Judge scores serve as our principal measure of correctness, with F1/B1 reported alongside to contextualize lexical fidelity.

**Latency (efficiency)**. We measure per-question retrieval latency. This involves the search process for memories (e.g., similarity computation, ranking). Additionally, we sum the retrieval latency time and the answer generation time in order to report a full end-to-end latency metric.

## 4.3 BASELINES

Our comparison strategy separates *breadth* and *depth* to provide clear, high-signal conclusions.

**LoCoMo (breadth).** LoCoMo's realistic two-speaker dialogues and rich category labels make it well-suited for a broad baseline matrix that teases apart design choices. We therefore compare against a diverse set of memory systems, including *Mem0* (API-based memory; Chhikara et al., 2025), *MemOS* (operating-system–style scheduler with MemCubes; Li et al., 2025), *LangMem* and *Zep* (commercial or open-source APIs; LangChain, 2024; Rasmussen et al., 2025), *RAG* (retrieval-augmented generation without persistent stores; Lewis et al., 2020), and a *full-context* control (upper bound with the entire conversation in context). This suite isolates the contributions of typed memory, minimal routing, and dense-only retrieval against widely used alternatives, enabling category-wise attribution of gains.

**LongMemEval (depth)**. LongMemEval serves as a generalization stress test rather than a leaderboard battleground. The guiding research question is:

*How does ENGRAM behave when conversational horizons expand by orders of magnitude?*

To answer this cleanly, we freeze the LoCoMo-validated configuration and compare only against a strong, architecture-agnostic, full-context control. This isolates horizon generalization effects while avoiding confounds from retriever engineering that a broad baseline panel would reintroduce. It also respects reproducibility constraints on a benchmark whose histories are already very long and heavy.

# 5 RESULTS

## 5.1 PERFORMANCE ON LOCOMO

To contextualize ENGRAM's performance, we evaluate against a diverse set of strong baselines spanning the principal design axes of previous state-of-the-art long-horizon memory systems (described in Section 4.3) while holding the LLM backbone fixed (`gpt-4o-mini`) to ensure consistency on the LoCoMo benchmark.

Table 1: **ENGRAM versus prior memory systems on the LoCoMo benchmark.** Reported metrics include LLM-as-Judge scores, token-level F1, and BLEU-1 (B1) across major QA categories and the overall aggregate.

| Category | Method | Chunk/Mem Tok | Top-K | LLM-as-Judge Score | F1 | B1 |
|----------|--------|---------------|-------|--------------------|-----|-----|
| single hop | langmem | 167 | – | $67.32 \pm 0.10$ | 41.74 | 34.82 |
| | mem0 | 1182 | 20 | $73.65 \pm 0.12$ | **46.26** | **40.54** |
| | memOS | 1596 | 20 | $78.32 \pm 0.16$ | 45.35 | 38.31 |
| | openai | 4104 | – | $61.73 \pm 0.24$ | 36.78 | 30.45 |
| | zep | 2301 | 20 | $51.42 \pm 0.17$ | 32.44 | 27.37 |
| | ENGRAM | 919 | 20 | **$79.90 \pm 0.12$** | 23.13 | 13.68 |
| multi hop | langmem | 188 | – | $56.71 \pm 0.12$ | **36.02** | **27.23** |
| | mem0 | 1160 | 20 | $57.85 \pm 0.20$ | 35.43 | 25.87 |
| | memOS | 1534 | 20 | $63.70 \pm 0.01$ | 35.37 | 26.56 |
| | openai | 3967 | – | $59.82 \pm 0.04$ | 33.02 | 23.18 |
| | zep | 2343 | 20 | $42.10 \pm 0.06$ | 23.11 | 14.69 |
| | ENGRAM | 919 | 20 | **$79.79 \pm 0.06$** | 18.32 | 13.23 |
| open domain | langmem | 211 | – | $49.56 \pm 0.16$ | **29.63** | **23.12** |
| | mem0 | 1151 | 20 | $44.93 \pm 0.02$ | 27.67 | 19.97 |
| | memOS | 1504 | 20 | $54.56 \pm 0.24$ | 29.46 | 22.32 |
| | openai | 4080 | – | $32.87 \pm 0.00$ | 17.17 | 11.01 |
| | zep | 2284 | 20 | $39.12 \pm 0.14$ | 19.98 | 13.67 |
| | ENGRAM | 895 | 20 | **$72.92 \pm 0.17$** | 8.56 | 5.47 |
| temporal reasoning | langmem | 138 | – | $24.21 \pm 0.21$ | 38.30 | 32.21 |
| | mem0 | 1185 | 20 | $53.34 \pm 0.33$ | 45.20 | 38.03 |
| | memOS | 1662 | 20 | **$72.68 \pm 0.16$** | **53.34** | **45.95** |
| | openai | 4042 | – | $29.26 \pm 0.02$ | 23.40 | 18.36 |
| | zep | 2302 | 20 | $19.47 \pm 0.31$ | 18.62 | 14.43 |
| | ENGRAM | 911 | 20 | $70.79 \pm 0.19$ | 21.90 | 14.74 |
| overall | langmem | 168 | – | $55.28 \pm 0.13$ | 39.22 | 32.16 |
| | mem0 | 1177 | 20 | $64.73 \pm 0.17$ | 42.90 | 36.05 |
| | memOS | 1593 | 20 | $72.99 \pm 0.14$ | **44.20** | **36.75** |
| | openai | 4064 | – | $52.81 \pm 0.14$ | 32.08 | 25.39 |
| | zep | 2308 | 20 | $42.29 \pm 0.18$ | 27.07 | 21.50 |
| | **ENGRAM** | 916 | 20 | **$77.55 \pm 0.13$** | 21.08 | 13.31 |

Table 1 reports category-wise and overall performance on LoCoMo. ENGRAM achieves the **highest overall semantic correctness**, with an LLM-as-Judge score of 77.55 under a shared backbone and prompt. By category, the largest margins appear on multi-hop (79.79) and open-domain (72.92), and ENGRAM also leads on single hop reasoning (79.90). For temporal-reasoning questions, performance is stronger on all baselines except for memOS (72.68).

Importantly, these accuracy gains come with a smaller evidence budget (reported as "Chunk / Mem Tok"), which on average is 916 tokens for ENGRAM. This is more than a **35% decrease in memory tokens** when compared to almost all of the other baselines, indicating that typed dense retrieval concentrates support into a compact context while preserving (and often improving) correctness. As anticipated in Section 4.2, F1/B1 solely reward surface-level overlap, not semantic accuracy. Despite leading LLM-as-Judge scores across all categories, we find that ENGRAM's lexical scores appear

lower because some responses are longer. We therefore include F1/B1 simply as complementary diagnostics and use the LLM-Judge as the principal correctness signal.

To better understand the impact of the specific ENGRAM architecture, we conduct an ablation that removes typed routing, collapsing all utterances into one undifferentiated store. As reported in Appendix A, Table 4, this configuration produces a noticeable decline in semantic correctness, with overall performance dropping to 46.56%. These results confirm that typed separation is not merely an architectural convenience but a key factor in concentrating relevant evidence and sustaining accuracy at long horizons.

## 5.2 LATENCY ANALYSIS

Table 2: **Latency comparison across baselines and ENGRAM on the LoCoMo dataset.** Latency is reported as median (p50) and 95th percentile (p95) in seconds for both search and total response time. Overall LLM-as-a-Judge (J) reflects mean $\pm$ stdev accuracy on the full evaluation set.

| Method | K | Search (s) | | Total (s) | | Overall J |
|---|---|---|---|---|---|---|
| | | p50 | p95 | p50 | p95 | |
| RAG, K=1 | 128 | 0.285 | 0.825 | 0.776 | 1.828 | $47.78 \pm 0.05$ |
| | 256 | 0.251 | 0.713 | 0.748 | 1.631 | $50.23 \pm 0.12$ |
| | 512 | 0.242 | 0.641 | 0.775 | 1.723 | $46.23 \pm 0.17$ |
| | 1024 | 0.242 | 0.721 | 0.823 | 1.961 | $41.02 \pm 0.06$ |
| | 2048 | 0.256 | 0.754 | 0.998 | 2.184 | $38.02 \pm 0.08$ |
| | 4096 | 0.258 | 0.719 | 1.096 | 2.714 | $36.09 \pm 0.11$ |
| | 8192 | 0.275 | 0.841 | 1.402 | 4.482 | $43.57 \pm 0.16$ |
| RAG, K=2 | 128 | 0.265 | 0.768 | 0.774 | 1.845 | $60.03 \pm 0.07$ |
| | 256 | 0.257 | 0.804 | 0.821 | 1.909 | $60.45 \pm 0.24$ |
| | 512 | 0.245 | 0.833 | 0.832 | 1.745 | $58.27 \pm 0.02$ |
| | 1024 | 0.234 | 0.862 | 0.861 | 1.880 | $50.34 \pm 0.19$ |
| | 2048 | 0.265 | 1.106 | 1.104 | 2.794 | $49.16 \pm 0.08$ |
| | 4096 | 0.271 | 1.461 | 1.461 | 4.832 | $51.82 \pm 0.13$ |
| | 8192 | 0.292 | 2.367 | 2.347 | 9.949 | $61.26 \pm 0.06$ |
| full-context | – | – | – | 9.940 | 17.832 | $72.60 \pm 0.07$ |
| langMem | – | 16.36 | 54.34 | 18.43 | 61.22 | $55.28 \pm 0.13$ |
| mem0 | 20 | **0.154** | **0.210** | 0.718 | 1.630 | $64.73 \pm 0.17$ |
| memOS | 20 | 1.806 | 1.983 | 4.965 | 7.957 | $72.99 \pm 0.14$ |
| openAI | – | – | – | **0.524** | **0.912** | $52.81 \pm 0.14$ |
| zep | 20 | 0.554 | 0.812 | 1.347 | 3.031 | $42.29 \pm 0.18$ |
| **ENGRAM** | 20 | 0.603 | 0.806 | 1.487 | 1.819 | $\mathbf{77.55 \pm 0.13}$ |

Table 2 shows that ENGRAM achieves both low latency and high semantic correctness on Lo-CoMo. Its median search and total times are 0.603 s and 1.487 s, alongside an LLM-as-Judge score of 77.55. Relative to full-context, which reports a median total of 9.940 s and 72.60 J, ENGRAM is significantly faster ($\approx$85%) while also outperforming in regards to accuracy. Because LoCoMo's dialogues generally fit within modern context windows, full-context is a strong, model-specific reference for "best case" access to history (though not a strict upper bound due to distraction and lost-in-the-middle effects), which makes ENGRAM's gains especially noteworthy.

## 5.3 TESTING WITH SCALE

We probe scaling behavior on LongMemEval$_S$ using the identical configuration validated on Lo-CoMo. As shown in Table 3, ENGRAM attains an overall LLM-as-Judge of 71.40%, surpassing the full-context control (56.20%) while consuming only $\approx$1.0–1.2K tokens per query versus 101K. This is approximately a 99% token reduction in input length. Similar compression-based strategies also preserve performance (Fei et al., 2023). Given that full-context affords the backbone maximal direct access to the dialogue, this comparison isolates the benefit of selective retrieval over indiscriminate inclusion: ENGRAM filters the history into a compact evidence set that the model can

Table 3: **Performance comparison on the LongMemEval benchmark.** We report accuracy across question types for GPT-4O-MINI, comparing a full-context baseline with ENGRAM. ENGRAM uses ≈99% fewer tokens while maintaining higher accuracy (71.40).

| Question Type | Full-context (101K tokens) | ENGRAM (1.0K-1.2K tokens) |
|---|---|---|
| single-session-preference | 23.33% | 93.33% |
| single-session-assistant | 92.86% | 87.50% |
| temporal-reasoning | 37.59% | 55.64% |
| multi-session | 39.10% | 60.15% |
| knowledge-update | 79.49% | 74.36% |
| single-session-user | 82.86% | 97.14% |
| **Overall J** | 56.20% | **71.40%** |

reliably act on, rather than relying on the model to sift through an order-of-magnitude larger prompt at inference time.

The qualitative implication is that ENGRAM's architectural bias, typed writes and dense set aggregation, acts as an effective information bottleneck at extreme horizons. Instead of amplifying "lost-in-the-middle" effects, the system consistently surfaces high-signal events, facts, and procedures that are causally relevant to the query. We view this as evidence that typed dense memory constitutes a scalable prior for long-horizon reasoning: the same design that produces competitive accuracy and latency on realistic, category-rich conversations also transfers to histories that are orders of magnitude longer, maintaining correctness while drastically reducing the contextual burden placed on the base model. This suggests that ENGRAM's architectural bias is not only effective for benchmarks, but also promising for deployment in real-world systems where long histories and strict efficiency constraints are the norm.

## 6 DISCUSSION

**Why a simple memory layer works.** Across both LoCoMo and LongMemEval$_S$, the ENGRAM rows in Tables 1, 2, and 3 consistently show that a compact design—typed memories, minimal routing, and dense-only retrieval with set aggregation—outperforms substantially more elaborate systems in both accuracy and efficiency. These results challenge the assumption that long-term memory requires increasingly complex schedulers or heavy graph construction (Li et al., 2024; 2025), and instead show that typed separation with straightforward retrieval is sufficient for state-of-the-art accuracy.

**Typed separation reduces competition at retrieval.** A key ingredient is the typed partitioning of memory into episodic, semantic, and procedural stores. By performing per-type top-$k$ retrieval and then merging with deduplication (Fig. 1; Section 3), ENGRAM limits cross-type competition and avoids pitting unrelated items against each other in a single global ranking. The results show especially strong gains on multi-hop and open-domain questions, where reasoning benefits from heterogeneous evidence: event timelines (episodic), stable facts (semantic), and instructions or protocols (procedural). Typed retrieval ensures that each evidence mode is represented before the final truncation step, rather than being washed out by a monolithic scorer.

**Accuracy–efficiency frontier.** The latency comparison in Table 2 shows that ENGRAM shifts the accuracy–efficiency frontier favorably. It achieves high semantic correctness while keeping response times low. This is not only a systems win; it is a modeling win. By constraining the prompt to a compact, high-signal subset, retrieval frees capacity for the answering model to reason, rather than forcing it to sift through long contexts that are prone to distraction and "lost-in-the-middle" effects (Liu et al., 2023; Wang et al., 2023).

**Token economy without accuracy loss.** On LoCoMo, ENGRAM operates with a smaller evidence budget than most baselines while still maintaining the strongest semantic correctness. On LongMemEval$_S$ it preserves accuracy while reducing tokens by roughly two orders of magnitude. This reduction improves throughput, lowers inference cost, and reduces variance from prompt-length interactions.

The results suggest several promising directions. First, *learning to route* could be guided by weak supervision from Judge-derived gradients or distillation from stronger backbones, complementing reinforcement-learning approaches to memory management (Yan et al., 2025). Second, *dynamic k and per-type budgets* may be tuned to query uncertainty. Third, *lightweight cross-type re-ranking* could preserve the set-aggregation discipline while capturing dependencies across stores. Fourth, *editable memory governance* opens space for user-facing controls such as redaction and temporal decay. Finally, *domain transfer* to areas like tutoring, customer support, and on-device assistants would stress efficiency under strict token and latency constraints. Across these directions, the guiding principle remains the same: preserve the simplicity that makes ENGRAM fast, reproducible, and easy to adopt.

## 7 CONCLUSION

In this work, we introduced ENGRAM, a compact memory architecture that couples typed extraction with minimal routing and dense-only retrieval via set aggregation. Despite its simplicity, ENGRAM delivers strong empirical performance: it achieves **state-of-the-art** semantic correctness on LoCoMo (**LLM-as-a-Judge 77.55%**) under a shared backbone and prompt (Table 1), pairs this accuracy with **low latency** (median total 1.487 s; Table 2), and generalizes to LongMemEval$_S$ where it **surpasses a strong full-context control** while using $\approx$**99% fewer** tokens (Table 3). These results indicate that careful memory typing and straightforward retrieval are sufficient to sustain long-horizon reasoning while improving efficiency.

**Limitations and future work.** Like any system, ENGRAM has constraints. Its effectiveness depends on dense retrieval quality, and catastrophic misses (e.g., paraphrased facts outside the embedder's neighborhood) can propagate directly to answers. The router is intentionally minimal, so complex utterances spanning multiple categories may require soft routing or overlapping writes. Our evaluation employs GPT-4o-mini as the judging model, and while we report mean $\pm$ standard deviation across runs, judge bias remains a limitation. The current formulation is also text-only and English-centric; extending to multilingual and multimodal settings will require type-aware encoders and language-specific extractors. Finally, typed separation improves interpretability but may underrepresent cross-type interactions that benefit from joint modeling (e.g., procedural steps conditioned on evolving episodic context). Approaches that fine-tune explicit memory interfaces offer a complementary path (Modarressi et al., 2024). We see promising directions in learned routing, adaptive per-type budgets, lightweight cross-type re-ranking, and extending memory governance to support editing, temporal decay, and privacy constraints.

Taken together, the ENGRAM rows across Tables 1, 2, and 3 tell a coherent story: a simple, typed memory layer can be both *accurate* and *efficient* at long horizons. In doing so, we directly challenge the prevailing trend toward increasingly complex graph schedulers and multi-stage pipelines, showing instead that careful memory typing and straightforward dense retrieval suffice to deliver state-of-the-art long-term consistency. We hope ENGRAM and its accompanying artifacts serve the community as a transparent, reproducible baseline and a principled foundation for advancing long-term memory in language models.

## REPRODUCIBILITY STATEMENT

We provide an anonymized repository at `https://anonymous.4open.science/r/engram-68FF/` containing code for the ENGRAM system and its evaluation, experiment scripts to reproduce all main results, and detailed setup files that specify datasets and parameters. Implementation specifics and additional analyses are documented in the appendix (including ablations in Appendix A and prompts in Appendix D). Together, these materials are intended to enable an independent reader to reproduce our results with ease and to encourage open discussion.

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

# A  APPENDIX A: ABLATIONS

## A.1  LOCOMO CATEGORY-WISE ABLATION AND NO TYPED ROUTING

Table 4: **Ablations by single-store variants.** Results on LoCoMo when restricting ENGRAM to single memory stores, independently testing each store's contribution to the system.

(a) Episodic-Only Store

| Category | LLM Score |
|---|---|
| single hop | 72.06% |
| multi hop | 61.70% |
| open domain | 45.83% |
| temporal reasoning | 67.60% |
| **overall** | 66.60% |

(b) Semantic-Only Store

| Category | LLM Score |
|---|---|
| single hop | 65.40% |
| multi hop | 58.87% |
| open domain | 41.67% |
| temporal reasoning | 59.81% |
| **overall** | 61.56% |

(c) Procedural-Only Store

| Category | LLM Score |
|---|---|
| single hop | 65.28% |
| multi hop | 53.19% |
| open domain | 47.92% |
| temporal reasoning | 32.09% |
| **overall** | 55.06% |

(d) Single Memory Store

| Category | LLM Score |
|---|---|
| single hop | 46.10% |
| multi hop | 35.51% |
| open domain | 33.33% |
| temporal reasoning | 52.44% |
| **overall** | 46.56% |

Collapsing episodic, semantic, and procedural stores into a single undifferentiated store (d) yields the weakest overall performance of **46.56%**, underscoring the value of typed separation (Table 4). Among single-store variants, the *Episodic-Only Store* performs best overall at **66.60%**, followed by the *Semantic-Only Store* at **61.56%**, and the *Procedural-Only Store* at **55.06%**. Independently, no single store comes within 10 absolute points of ENGRAM's overall score of **77.55%** on LoCoMo.

## A.2  K-VALUE SCALING VS ACCURACY

We scale the final retrieval budget hyperparameter $K$ and observe an accuracy–efficiency trade-off.

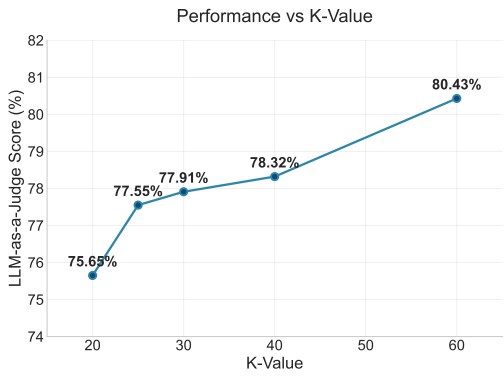

(a) Accuracy increases with $K$ (knee at $K \approx 25$)

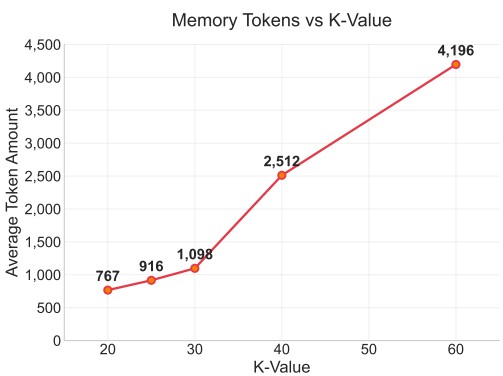

(b) Context cost grows steeply beyond $K=30$

As $K$ increases from 20 to 60, the LLM-as-a-Judge (J) score rises monotonically from **75.65** $\rightarrow$ **80.43** (left). However, the average token budget of retrieved evidence grows from **767** $\rightarrow$ **4196** (right), a **4.6×** increase. The knee of the curve is at $K \approx 25$: relative to $K=20$, moving to $K=25$ yields a sizable +1.90 J for only 149 tokens, while $K=30$ adds just 0.36 J for +182 tokens. Beyond $K=40$, gains are modest per token: $30 \rightarrow 40$ improves +0.41 J but costs +1414 tokens; $40 \rightarrow 60$ recovers +2.11 J at +1684 tokens.

To quantify marginal utility, we compute improvement per 1k tokens added: **12.8** (20→25), **1.98** (25→30), **0.29** (30→40), and **1.25** (40→60) J-points/1k tokens. Thus, $K{=}25$ maximizes return on context, justifying our choice for $K$ in Section 3. If an application prioritizes absolute peak accuracy and can afford a $\approx$4–5$\times$ larger evidence window, $K{=}60$ is best; otherwise **K$=$25** has the strongest accuracy-to-cost balance.

# B  APPENDIX B: DATASETS

## B.1  LOCOMO

**Problem setting.** LOCOMO consists of long-term, multi-session dialogues between two speakers. Conversations are constructed via a human–machine pipeline grounded in personas and event graphs, followed by human editing for long-range consistency. Each dialogue is accompanied by a large set of probing QA items that assess whether systems can retrieve and compose information spread across distant turns.

**Scale and structure.** The released benchmark contains **10 dialogues**, each averaging ≈**600 turns** and ≈**16K tokens**, distributed over **up to 32 sessions**. Each turn is speaker-attributed, enabling per-speaker memory construction and retrieval. This design stresses attribution (who said what, when) in addition to content recall.

**QA categories.** Questions are labeled into five categories: *single-hop*, *multi-hop*, *temporal*, *commonsense/world knowledge*, and *adversarial/unanswerable*. Following common practice in the literature and our main text, we report metrics on the first four categories and exclude adversarial/unanswerable items from quantitative aggregates (retaining them for qualitative error analysis).

**Evaluation notes.** In our setup, we (i) isolate memory per dialogue to avoid cross-example leakage, (ii) maintain distinct stores per speaker, and (iii) evaluate each QA item independently with a fixed answering prompt and deterministic decoding. We report semantic-correctness (LLM-as-a-Judge) as the primary metric, with F1/BLEU as lexical complements, and latency for search and end-to-end response.

## B.2  LONGMEMEVAL

**Problem setting.** LONGMEMEVAL embeds curated QA tasks within long, synthetic user–assistant conversations designed to stress-test memory. Each history is constructed to be substantially longer than those in LoCoMo, providing extended contexts in which systems must retrieve, update, and reason over information. Beyond straightforward recall, the benchmark emphasizes robustness at scale: questions often depend on temporally ordered updates, require correct handling of multi-session histories, and include cases where the system must abstain when no sufficient evidence exists.

**Scale and split.** The benchmark provides **500 curated questions** embedded in length-configurable chat histories. In this work we use the LONGMEMEVAL$_S$ scale, which comprises histories of approximately **115K tokens per problem**. This scale exceeds that of LoCoMo by over 7x, providing a strong scaling measure for stress testing ENGRAM.

**Abilities.** Questions are organized into five evaluation abilities: *information extraction*, *multi-session reasoning*, *temporal reasoning*, *knowledge updates*, and *abstention*. The abstention ability is treated as a decision problem rather than a span-matching problem; span-based lexical metrics are not computed on abstention items.

**Evaluation notes.** We port the configuration validated on LOCOMO (typed writes, dense-only retrieval, fixed evidence budget, identical answering prompt) without retuning, and compare to a strong full-context control. We report ability-wise metrics, overall semantic correctness, and latency as a function of total history length.

## C    APPENDIX C: END-TO-END ENGRAM QA DIAGRAM

### C.1    FULL QUESTION-ANSWER WALKTHROUGH

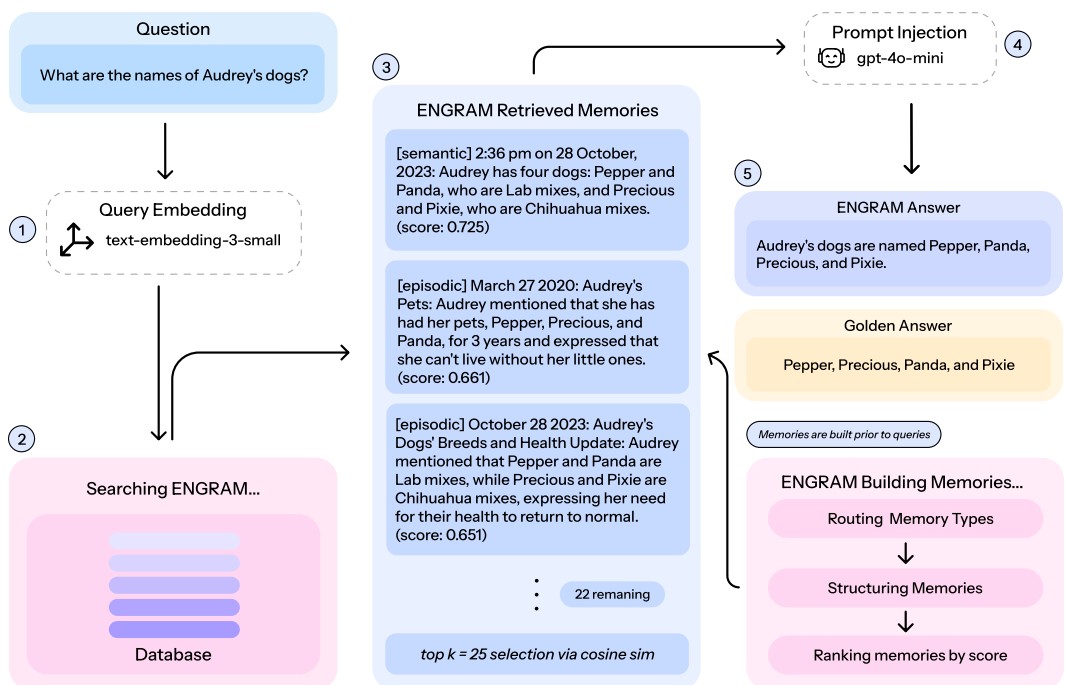

Figure 3: **End-to-end ENGRAM QA walkthrough.** The diagram illustrates how turns are first routed into typed stores (episodic/semantic/procedural), normalized, and embedded then—at query time—how the query is embedded and used to retrieve per-type top-$k$ neighbors by cosine similarity (default $K=25$), followed by aggregation and deduplication. Finally, retrieved, timestamped records are serialized into a fixed prompt template and passed to the answering model (`gpt-4o-mini`). The figure also shows the concrete example ("What are the names of Audrey's dogs?"), the evidence snippets with similarity scores, and ENGRAM 's answer vs. gold answer. Embedding and model components (`text-embedding-3-small`, `gpt-4o-mini`) are annotated to emphasize the separation between memory construction, retrieval, and answer generation.

# D  APPENDIX D: PROMPTS

## D.1  ROUTER PROMPT

```
Given this message, determine which storage types are most relevant:

Message: {message}

Storage types:
    • episodic: Event and experiences with temporal context (think
      of this as a personal diary of events that is timelined)
    • semantic: Facts, observations, preferences (only route to this
      if the message reveals a non-event fact or preference about a
      person, place, or thing)
    • procedural: How-to information, processes

Determine which types are relevant for this message.
```

## D.2  EPISODIC MEMORY PROMPT

```
You are an intelligent memory assistant tasked with converting a
single conversation message into an episodic memory JSON object.

Context
You receive a raw message string and a conversation timestamp.  The
conversation timestamp reflects when the message was sent, not
necessarily when the event occurred.  You must infer the actual event
time from the message content and conversation context.

Instructions
    1. Preserve specific nouns exactly as written:  use exact place
       names, object/activity names, proper nouns, numbers, and titles
       verbatim.
    2. Reason carefully about time.  Do not blindly copy the input
       timestamp; infer the event timestamp from the message and
       context.
    3. Write in third person.
    4. Do not use temporal words in the summary (e.g., ``yesterday'',
       ``last week'').
    5. Return only a single JSON object with keys title, summary, and
       timestamp.

Approach (think step by step)
    1. Read the message and identify specific entities (names, places,
       objects) to preserve verbatim.
    2. Extract temporal cues (e.g., ``yesterday'', ``last month'',
       holiday names) and map them to absolute dates based on the
       conversation timestamp.
    3. If the message implies a date range (e.g., a month without a
       day), output the most specific resolvable form (e.g., ``June
       2024'').
    4. Form a concise title that includes specific details; then
       write a one-sentence summary in third person with exact nouns
       preserved.
    5. Compute the timestamp as the actual event date/time derived
       from the message (not merely the conversation time).

Timestamp reasoning examples
```

- Input timestamp: ``10:55 am on 22 July, 2024''; message: ``I went to Cheesequake park yesterday with my friends'' ⇒ event timestamp: ``July 21 2024''.
- Input timestamp: ``10:55 am on 22 July, 2024''; message: ``I went camping in Banff last month'' ⇒ event timestamp: ``June 2024''.

```
Inputs
Message: {message}
Conversation Timestamp: {timestamp}

Output (return only this JSON)
{"title": "...", "summary": "...", "timestamp": "..."}
```

## D.3 SEMANTIC MEMORY PROMPT

```
Extract ONE key fact or observation from this message.

CRITICAL: Preserve specific nouns exactly as mentioned.  Do NOT
generalize or paraphrase:
      • Use exact place names, not generic terms
      • Use exact object/activity names, not categories
      • Use exact relationship terms, not synonyms
      • Keep proper nouns, numbers, and specific terms verbatim

RULES:
      • One sentence with specific details preserved exactly as written
      • Include exact names, places, objects, activities as mentioned
      • Proper nouns must be preserved verbatim
      • Be specific and detailed, avoid generic terms

Message: {message}

Return only a JSON object:
{"fact": "detailed fact with specific nouns preserved exactly"}
```

## D.4 PROCEDURAL MEMORY PROMPT

```
Convert this message into procedural memory format:

Inputs
Message: {message}

Extract
      • title:  A title for this procedure/instruction
      • content:  The procedural content/steps (can be a string or list
        of steps)
```

## D.5 ANSWER GENERATION PROMPT

```
You are an intelligent memory assistant tasked with retrieving
accurate information from conversation memories.

Context
You have access to memories from two speakers in a conversation.
These memories contain timestamped information that may be relevant
to answering the question.
```

```
Instructions
    1. Carefully analyze all provided memories from both speakers.
    2. Pay special attention to the timestamps to determine the
       answer.
    3. If the question asks about a specific event or fact, look for
       direct evidence in the memories.
    4. If the memories contain contradictory information, prioritize
       the most recent memory.
    5. Always convert relative time references to specific dates,
       months, or years.  For example, convert ``last year'' to
       ``2022'' or ``two months ago'' to ``March 2023'' based on the
       memory timestamp.  Ignore the original relative phrasing when
       answering.
    6. Focus only on the content of the memories from both speakers.
       Do not confuse character names mentioned in memories with the
       actual users who created those memories.
    7. Keep the answer concise but preserve nouns exactly as they
       appear in the memories (e.g., use the specific food names
       given).

Approach (think step by step)
    1. Examine all memories that relate to the question.
    2. Check timestamps and content carefully.
    3. Identify explicit mentions of dates, times, locations, or
       events that answer the question.
    4. If conversion is needed (e.g., relative → absolute dates),
       briefly show the calculation.
    5. Formulate a precise, concise answer based solely on evidence in
       the memories.
    6. Double-check that the answer directly addresses the question.
    7. Ensure the final answer uses specific time references (no vague
       phrases).

Inputs
Memories for user {speaker_1_user_id} {speaker_1_memories}
Memories for user {speaker_2_user_id} {speaker_2_memories}

Question:  {question}
```