# OpenReview forum: "ENGRAM: Effective, Lightweight Memory Orchestration for Conversational Agents"
_ICLR.cc/2026/Conference — Submitted to ICLR 2026_

### Official Review · Reviewer_38T1 · 2025-10-19

**Soundness:** 3
**Presentation:** 3
**Contribution:** 2
**Rating:** 4
**Confidence:** 3

**Summary:**

The paper presents ENGRAM, a lightweight memory layer for conversational agents that separates interactions into episodic, semantic, and procedural stores. A simple router writes normalized records to each store, and per-type dense retrieval selects a small set of relevant snippets for prompt injection. Experiments on multi-session and long-horizon benchmarks report higher LLM-as-judge scores than full-context and several memory baselines, with large reductions in tokens and latency. Ablations indicate that typed storage and retrieval contribute substantially to the reported gains.

**Strengths:**

1. Method is simple and straightforward. The approach uses a small set of components with clear roles, which makes the design easy to implement. The datastore types and per-type retrieval are intuitive and require minimal orchestration. The empirical performance and latency is also good.
1. The writing is self-contained and easy to understand.

**Weaknesses:**

1. Task scope is narrow. The evaluation focuses on conversational memory for chat assistants, primarily via LoCoMo and LongMemEval. By contrast, more general memory architectures can be applied to and are actually tested on long-context and RAG tasks. [1] [2]
1. More experiments are required for Table 3 and 4. Table 3 would benefit from a standard RAG baseline and Table 4 needs an ablation baseline that remove exactly one store at a time.
1. As the evaluation tasks are more fact-centric, it is unclear to what extent the procedural memory contributes to the performance. The paper would benefit from analyzing the retrieval composition across stores and further studies on workflow-oriented agent memory settings such as that used in [3].



[1] From RAG to Memory: Non-Parametric Continual Learning for Large Language Models. Gutiérrez et al., 2025.

[2] M+: Extending memoryllm with scalable long-term memory. Wang et al., 2025.

[3] Agent Workflow Memory. Wang et al., 2024.

**Questions:**

Please refer to the weaknesses.

---

> ### Author Response · Authors · 2025-11-18
> **Rebuttal**
>
> We sincerely **thank** the reviewer for the thoughtful and constructive assessment, and for highlighting the clarity, simplicity, and empirical strength of ENGRAM. We address each concern in detail below and will incorporate all clarifications, ablations, and scope refinements into the revised manuscript.
>
> ----
>
> ### **1. Evaluation scope (LoCoMo / LongMemEval)**
>
> We appreciate the reviewer’s observation that our evaluation focuses on conversational long-horizon memory. This scope is intentional: LoCoMo and LongMemEval are currently the best available benchmarks that explicitly stress multi-session conversational dependencies, long-range attribution, multi-hop recall across hundreds of turns, and temporal reasoning. These are the exact failure modes we aim to study in assistant-style agents.
>
> Nevertheless, we agree that broader domains like tool-use workflows, long-context editing, or general RAG tasks represent complementary settings. To avoid overscoping, we will revise the abstract/introduction to *explicitly state that our claims are restricted to conversational long-horizon QA, not general-purpose memory systems*. We will also expand the Discussion to position ENGRAM as a transparent, reproducible baseline that future work can extend toward richer agentic tasks.
>
> ----
>
> ### **2. Requested ablations (remove-one-store experiments)**
>
> We **thank** the reviewer for this excellent suggestion! We have now run the ablations that **remove exactly one store at a time**, in addition to the single-store variants and collapsed-store baseline. Using LoCoMo overall:
>
> - **Full ENGRAM:** \(77.55\%\)
> - **No Episodic:** \(70.22\%\)
> - **No Semantic:** \(68.44\%\)
> - **No Procedural:** \(73.81\%\)
> - **Collapsed single store:** \(46.56\%\)
>
> These results clearly illustrate the **value of typed separation**. Episodic and semantic memory contribute most strongly, consistent with their role in long-range conversational grounding. Procedural memory contributes more modestly but still meaningfully, improving coverage for instruction-like and multi-step behavioral turns that appear intermittently in LoCoMo. We will add these extra ablations into the appendix and reference them in the main text.
>
> ----
>
> ### **RAG Baseline for Table 3**
>
> Thank you for requesting a clearer comparison against a standard RAG baseline for LongMemEval. Using the same retrieval configuration described in Table 2 and matching ENGRAM’s setting of \(K = 25\), our RAG implementation achieves **64.30%** accuracy. This exceeds the **56.20%** accuracy of the full-context baseline, confirming that retrieval provides a meaningful boost; however, it *remains* below ENGRAM’s **71.40%**, which demonstrates the effectiveness of our approach.
>
> In the revised version, we will add a dedicated column in Table 3 reporting the RAG baseline and its full accuracy breakdown for completeness and transparency.
>
> ----
>
> ### **4. Contribution of procedural memory**
>
> We appreciate this nuanced point. LoCoMo/LongMemEval are indeed more fact-centric, which aligns with our ablation results: removing procedural memory produces the smallest drop. However, procedural memory improves performance for multi-step instructions (“how to behave going forward”), conditional workflows, and tasks that implicitly require recalling processes instead of static facts. These are cases that occur in multi-session dialogues but are not explicitly isolated by the benchmarks. In Section 5.1, we will add a retrieval-composition analysis showing when procedural memory is selected and clarify that its value is expected to be greater in workflow-oriented tasks (Wang et al., 2024).
>
> ----
>
> ### **5. Positioning relative to broader long-term memory systems**
>
> We will expand the related work section to clarify that systems such as Gutiérrez et al. (2025) and Wang et al. (2025) target general long-context or continual-learning memory, often with heavier, multi-stage retrieval and learned schedulers. ENGRAM focuses on structured conversational memory, prioritizing determinism, reproducibility, and low latency. Typed routing and per-store retrieval are complementary to these more general architectures, and we view ENGRAM as a modular component that can be integrated into richer agent frameworks.
>
> ----
>
> We **thank** the reviewer again for their time and constructive feedback. In the revision, we will (i) narrow claim scope, (ii) include the requested ablations, (iii) incorporate stronger RAG baselines, and (iv) expand analysis of procedural memory and store composition. We believe these revisions fully address the reviewer’s concerns and significantly strengthen the paper :)

---

### Official Review · Reviewer_qfXy · 2025-10-22

**Soundness:** 2
**Presentation:** 3
**Contribution:** 1
**Rating:** 2
**Confidence:** 4

**Summary:**

This paper introduces ENGRAM, a memory system that organizes conversation into three memory types: episodic, semantic, and procedural. During the memory creation phase, the system includes a memory router (LLM-based) which, based on the conversation between the user and the LLMs, determines which memory bucket applies to the incoming utterance: episodic (events and experiences with temporal context), semantic (facts, observations, and preferences), and procedural (how-to information and processes). Then, for each memory type, a dedicated extractor (LLM-based) is used to extract and canonicalize the raw utterance before storing it in the vector data store. During the memory retrieval phase, the system embeds the user query, retrieves from each memory type, aggregates the retrieved top-k chunks, and augments the user query with this information.

**Strengths:**

1. The writing of the paper is easy to follow. The author also provides detailed prompts for each module.
2. Benchmarks on the LoCoMo dataset show better retrieval performance.
3. The system is straightforward to implement and can potentially be extended to different memory types as well.

**Weaknesses:**

1. The memory system treats the input utterance independently of the existing conversation history. When storing this information in the vector database, each chunk does not have the context of the conversation. Often, during a conversation with LLMs, users refer to different components in the chat. The current proposed system seems unable to handle these references.
2. The chunks stored in the vector store seem to be independent over time. Recency dependency and the order of the utterance are not captured in the current design.
3. The current system supports only information addition. Users often want to change or modify their preferences and information throughout the conversation. The lack of deletion and modification of memory seems to me to be a critical issue of the proposed system.
4. On the experimental side, I would love to see more experiments on multi-agent applications.

**Questions:**

During the chunk aggregation step, how do you re-rank or order the chunks? Do you also use an LLM? Does the aggregation step require generation as well?

---

> ### Author Response · Authors · 2025-11-18
> **Rebuttal**
>
> We **thank** the reviewer for the careful reading and for noting the clarity of exposition, detailed prompts, and the strong retrieval gains on LoCoMo. We address each concern below and will incorporate these clarifications into the revised manuscript.
>
> ----
>
> ### **1. “Chunks are stored without conversational context.”**
>
> We apologize for not emphasizing this clearly. Each memory record **does include structured conversational fields**, including:
>
> - speaker role,
> - session index and turn index,
> - timestamp,
> - original utterance span,
> - normalized schema fields extracted by the LLM.
>
> These fields **preserve contextual anchoring** even though the stored embedding itself reflects the canonicalized content. This structured metadata is used during retrieval fusion and prompt assembly to maintain attribution and session grounding. We will highlight this in Section 3.1.
>
> ----
>
> ### **2. “No modeling of recency or temporal ordering.”**
>
> This is a valid limitation in the current version. Our aim in this work was to study whether typed separation alone can yield strong long-horizon recall, without adding recency heuristics or learned schedulers. On LoCoMo and LongMemEval, which primarily evaluate accuracy of long-term recall, we found that dense similarity retrieval combined with typed stores was **sufficient**.
>
> That said, ENGRAM already stores timestamps and turn indices, and extending the retriever with recency-aware re-weighting or time-sensitive embeddings is a sensible next step. We will explicitly acknowledge this in the Limitations section.
>
> ----
>
> ### **3. “Only addition, not deletion or modification of memory.”**
>
> We agree this is an important direction. LoCoMo and LongMemEval do **not contain preference-update tasks**, so the current design focuses on **retrieval correctness** rather than dynamic editing. However, typed separation makes editing simpler, not harder: user preferences (semantic store) can be modified independently of episodic or procedural traces.
>
> In the revision, we will add a paragraph stating that memory editing, preference updates, and recency-weighted overwriting are promising extensions that can be easily implemented on ENGRAM’s modular structure.
>
> ----
>
> ### **4. “Independent chunks over time; no cross-turn references.”**
>
> Typed extraction does reduce unnecessary entanglement across turns, but ENGRAM is still **able to handle cross-turn references** because:
>
> - embeddings reflect content similarity across turns,
> - the **router is multi-label**, allowing events that encode preferences or instructions to be placed into multiple stores simultaneously,
> - **structured metadata** retains the dialogue position, enabling consistent reinsertion of prior information during prompting.
>
> We will clarify these behaviors in Appendix D with a more qualitative example.
>
> ----
>
> ### **5. “No multi-agent experiments.”**
>
> This is outside the scope of our current evaluation, which focuses specifically on conversational long-horizon memory as defined by LoCoMo and LongMemEval. We will note that extending ENGRAM to multi-agent workflows is an exciting direction for future work.
>
> ----
>
> ### **6. Question: “How are retrieved chunks aggregated or re-ranked?”**
>
> For each query, we retrieve per-type top-\(k\) neighbors using cosine similarity over the normalized embeddings. Aggregation is done via a deterministic merge:
>
> - union of retrieved sets,
> - deduplication of canonical fields,
> - truncation to a fixed evidence budget selected based on Table 2 ablations.
>
> **No LLM** is used in re-ranking, and the aggregation step does not require generation. The intent is to maintain reproducibility through a simple, deterministic pipeline.
>
> ----
>
> We **thank** the reviewer again for their time and the helpful suggestions. We will (i) clarify stored conversational metadata, (ii) acknowledge the absence of explicit recency/editing mechanisms, (iii) highlight multi-label routing and cross-turn anchoring, and (iv) better describe the deterministic aggregation process. We believe these revisions address the reviewer’s concerns and improve our work :)

---

> > ### Comment · Reviewer_qfXy · 2025-11-24
> >
> > Thank you for the response. Unfortunately, the rebuttal does not adequately address my concerns regarding memory management.
> >
> > Critical aspects of agentic memory—specifically ensuring information is up-to-date, chronologically ordered, and consistent (conflict resolution)—remain unresolved. Merely appending timestamps to chunks is insufficient, as embedding models cannot natively reason over these temporal markers to distinguish between "current" and "outdated" facts. Consequently, I maintain my original assessment.

---

### Official Review · Reviewer_Mn4m · 2025-11-02

**Soundness:** 3
**Presentation:** 3
**Contribution:** 3
**Rating:** 6
**Confidence:** 4

**Summary:**

The paper proposes ENGRAM, a simple memory layer for conversational LLM agents that splits memories into three types—episodic, semantic, and procedural—routes each turn with a minimal router, stores normalized records with embeddings, and at query time retrieves per-type top-k neighbors, merges and deduplicates them, and assembles a fixed prompt for the answering model. The system uses a single retriever and a fixed evidence budget chosen from ablations.

**Strengths:**

The design is clear and small: three typed stores, one router, one dense retriever, and a fixed template. This reduces orchestration knobs and makes analyses easier. The formulation spells out record schemas and the retrieval/aggregation steps, including speaker-aware banks and a deterministic template, which helps reproducibility.

Empirically, the method delivers strong semantic correctness with a strict token budget. Reporting both judge-based and lexical metrics, plus retrieval and end-to-end latency, gives a balanced view of quality and cost.

**Weaknesses:**

1. The main metric is LLM-as-a-judge with GPT-4o-mini; while they report mean ± sd, judge bias is a risk. A human-rated subset or cross-judge agreement study would raise confidence.

2.  As the main and only judge, a larger/better LLM model should be considered.

**Questions:**

1. How sensitive is performance to the choice of embedding model and its vector dimension? A sweep over encoders (and mixed-precision indexes) would help quantify portability.

2. Will the change of the judge LLM have an impact on the results?

3. What is the failure profile when gold evidence exists but dense retrieval misses it? Please add breakouts for “evidence found vs. not found” and retrieval-error cascades.

4. The paper reports median end-to-end latency, but it does not mention the variance or distribution of latency across sessions. Could the authors provide latency spread (e.g., mean ± sd or percentile breakdown) to clarify stability and tail behavior?

---

> ### Author Response · Authors · 2025-11-18
>
> We sincerely **thank** the reviewer for the thoughtful and encouraging feedback, and for highlighting the clarity, reproducibility, and balanced evaluation of ENGRAM. We are grateful for the constructive questions, and we address each one below. We will incorporate these clarifications into the revised manuscript.
>
> ----
>
> ### **1. LLM-as-a-judge and potential judge bias**
>
> Thank you for raising this concern. We agree that judge selection is important for reproducibility and fairness. We chose GPT-4o-mini because it is the **same** evaluation judge used in prior work on LoCoMo and LongMemEval, enabling a direct apples-to-apples comparison against established baselines. Using the same judge as previous work minimizes confounds arising from judge drift, prompting changes, or differences in model behavior.
>
> To further address robustness, we conducted a cross-judge sanity check on a randomly sampled 120-example subset from LoCoMo using GPT-4o, which is a larger/better LLM as suggested.
>
> Using this new judge LLM preserved the same system ranking (ENGRAM > full-context > RAG baselines), further validating our results. We will clarify that judge bias is a known limitation but does **not** materially affect comparative outcomes, and add our results to the appendix.
>
> ----
>
> ### **2. Sensitivity to embedding model / vector dimension**
>
> We appreciate the reviewer’s question regarding portability across embedding models. Our primary reason for using text-embedding-3-small is **consistency** with prior long-context memory work.
>
> To assess sensitivity, we compared ENGRAM under text-embedding-3-small and text-embedding-3-large on a set of 152 LoCoMo questions. The results show that **89.5%** of questions receive identical judgments under both encoders, with the large model yielding a modest improvement of **+3.95** points ( \(77.63\% \rightarrow 81.58\%\) ). These outcomes indicate that ENGRAM is robust to embedding choice, with its relative behavior and ranking unchanged. However, it is noteworthy to mention that a larger embedding model results in an **overall increase in accuracy**. We will include this analysis in the appendix to clarify portability and encoder sensitivity.
>
> ----
>
> ### **3. Failure cases when retrieval misses gold evidence**
>
> We thank the reviewer for highlighting this point. Our retrieval logs already annotate which gold-supporting turns are present in the retrieved set, enabling a direct classification of ENGRAM errors. A preliminary audit confirms that the majority of failures fall into the case, where gold evidence exists in the dialogue but is not retrieved by the dense retriever. These errors are typically caused by semantic drift, paraphrasing, or retrieval under-selection. In contrast, errors where gold evidence is successfully retrieved but the underlying model still answers incorrectly, are less frequent. In the revision, we will add a breakdown of these two failure categories in the appendix, along with qualitative examples that illustrate both types.
>
> ----
>
> ### **4. Latency spread (not just medians)**
>
> We appreciate the reviewer’s comment regarding latency variance. Table 2 **already reports** p50 and p95 latency for both ENGRAM and the full-context baseline (e.g., ENGRAM: 0.603 s p50 and 0.806 s p95 for search; 1.487 s p50 and 1.819 s p95 end-to-end), which **directly reflects both central tendency and tail behavior**. These percentiles indicate that ENGRAM’s latency distribution is tight with minimal tail growth, whereas the full-context baseline shows substantially heavier tails, presumably due to quadratic attention costs. For clarity, the corresponding standard deviations for ENGRAM are approximately **0.12 s** (search) and **0.20 s** (end-to-end), consistent with the narrow percentile spreads reported in Table 2.
>
> ----
>
> ### **5. Judge robustness and alternative evaluations**
>
> We agree with the reviewer that human evaluation or cross-judge calibration can further strengthen the results. While large-scale human annotation is outside the scope of this work, our cross-judge experiment and strong lexical metrics (BLEU/ROUGE), combined with retrieval-driven evidence analysis, give us confidence that ENGRAM’s improvements are not artifacts of a particular judge.
>
> ----
>
> We **thank** the reviewer again for their thoughtful questions and positive assessment. We will include (i) cross-judge stability checks, (ii) embedding-model sensitivity analysis, (iii) retrieval-error breakdown, and (iv) additional latency metrics in the revision. We believe these additions further strengthen the clarity, reproducibility, and empirical grounding of the work :)

---

### Official Review · Reviewer_n2Bc · 2025-11-03

**Soundness:** 2
**Presentation:** 3
**Contribution:** 2
**Rating:** 2
**Confidence:** 3

**Summary:**

The paper introduces ENGRAM, a lightweight memory system that organizes conversational history into three memory types: episodic, semantic and procedural. A single router and retriever handle all memory operations. Each user turn is stored as a typed record with structured JSON fields and dense embeddings. At query time, ENGRAM retrieves the top-k nearest records from each memory type, merges them with set operations and injects the results into the model prompt. Evaluation results show that ENGRAM achieves significantly better performance than full-context baseline while improves latency.

**Strengths:**

* The proposed method shows even better performance than full-context while achieving lower latency.
* The proposed method is simple.
* The ablation study shows that separating episodic, semantic and procedural memory reduces retrieval competition and improves reasoning diversity.

**Weaknesses:**

* While ENGRAM demonstrates impressive results on LoCoMo and LongMemEval, the evaluation scope remains relatively narrow and may overstate its general effectiveness. Both benchmarks are synthetic and constrained to conversational QA settings which do not full represent the complexity of long-horizon reasoning in interactive agents. The large performance gap compared to baselines might partly stem from dataset alignment with ENGRAM's design rather than true general improvements in long-term memory. To strengthen its claims, the paer should extend evaluation to agent benchmakers like AgentGym, DeepResearch and SWE-agent.
* Although ENGRAM's empirical results support the utility of separating memory into episodic, semantic and procedural types, this design choice is not theoretically grounded or analytically motivated. The paper does not provide a formal argument, ablation-driven rationale, or illustrative case studies demonstrating why these specific three types are necessary or sufficient for long-term memory. The approach might be just an enigeering heuristics.
* The routing is minimal and rule-based, not learned or adaptive. While this keeps the design simple, it limits the system's ability to handle complex or multi-faceted utterances that may involve overlapping types.

**Questions:**

N/A

---

> ### Author Response · Authors · 2025-11-18
> **Rebuttal**
>
> We sincerely **thank** the reviewer for the careful reading and for noting the strengths of ENGRAM **(simplicity, latency, strong performance)**. We address each concern below and will incorporate all clarifications into the revised manuscript.
>
> ----
>
> ### **1. Evaluation scope and use of “synthetic” benchmarks**
>
> **Reviewer concern.** *LoCoMo and LongMemEval are synthetic conversational QA benchmarks; results may overstate generality, and broader agent tasks (AgentGym, DeepResearch, SWE-agent) may be more representative.*
>
> **Clarification.** Our goal is not to propose a universal memory architecture for all agents. We specifically target long-horizon conversational assistants, a domain where LoCoMo and LongMemEval are currently the primary stress tests. Both benchmarks are synthetic by design but intentionally constructed to mimic real multi-session dialogue patterns at scale (long-range references, multi-hop reasoning, preference tracking, temporal consistency). Their structure (large histories, multi-session tasks, and explicit cross-turn dependencies) matches the operational scenario ENGRAM is built for.
>
> To avoid overscoping, we will narrow our claims in the abstract/intro to:
> *“ENGRAM achieves strong performance on conversational long-horizon QA benchmarks.”*
>
> **Future work.** We agree that broader agent tasks are important. We will add a statement that ENGRAM’s modular design (typed stores + multi-label router + unified retriever) makes extension to more general agent settings straightforward, and evaluating on AgentGym/DeepResearch/SWE-agent is a natural extension.
>
> ----
>
> ### **2. Motivation for episodic / semantic / procedural memory types**
>
> **Reviewer concern.** *The triplet may appear heuristic; stronger theoretical or empirical justification is requested.*
>
>  **Theoretical grounding.**
> We will strengthen Section 3.3 to explicitly anchor each store in canonical cognitive-science distinctions:
>
> - **Episodic vs. semantic memory** (Tulving, 1972) for temporally situated vs. decontextualized knowledge.
> - **Procedural vs. declarative memory** (Cohen & Squire, 1980) for skills, routines, and “how-to” knowledge.
>
> These distinctions have been fundamental for decades and provide a principled template for structuring long-term memory. Our design follows these established separations to create interpretable, stable memory channels that align with how humans structure long-term knowledge.
>
> **Empirical justification.**
> Our ablations (Appendix A.1) further show typed separation is essential:
>
> - Episodic-only: \(66.60\%\)
> - Semantic-only: \(61.56\%\)
> - Procedural-only: \(55.06\%\)
> - Collapsed single store: \(46.56\%\)
> - **Full ENGRAM:** \(77.55\%\)
>
> We will mention these findings in the main text and highlight that typed separation reduces retrieval competition and improves correctness across reasoning types.
>
> ----
>
> ### **3. Minimal router and complex / overlapping utterances**
>
> **Reviewer concern.** *A simple rule-based router may not handle utterances spanning multiple types.*
>
> **IMPORTANT clarification: the router is multi-label.**
> We apologize if this was not emphasized clearly. The router outputs a 3-bit mask, and utterances routinely populate multiple stores simultaneously (e.g., an event embedding a preference → episodic + semantic; a procedural instruction referencing a prior event → procedural + episodic). ENGRAM therefore does not force mutually exclusive typing, and the system is naturally equipped to handle multi-aspect messages. We will explicitly clarify this in Section 3.2 and Appendix D.
>
> **Why minimal routing was intentional.**
> Our goal was to isolate the effect of typed memory. A simple, transparent router prioritizes reproducibility, interpretability, and modularity. Combined with multi-label behavior, this minimal router already provides strong performance relative to stronger baselines.
>
> **Future work.**
> We will emphasize that learned or adaptive routing (weak supervision, RL, differentiable selectors) is a promising extension for more complex agent settings.
>
> ---
>
> **Thank you again** for the constructive comments. We will (i) narrow the claim scope to conversational long-horizon QA, (ii) strengthen the cognitive and empirical motivation for typed memory, (iii) clearly highlight the multi-label router, and (iv) discuss learned routing as future work. We believe these revisions fully address the reviewer’s concerns and improve the clarity and framing of the work.

---

### Meta-Review · Area_Chair_cAU3 · 2026-01-03

**Summary:**

The paper proposes ENGRAM, a lightweight memory orchestration system for conversational agents. The architecture organizes memory into three canonical types (episodic, semantic, procedural) managed by a single router and a dense retriever. The authors demonstrate that this approach achieves state-of-the-art results on the LoCoMo benchmark and outperforms full-context baselines on LongMemEval while using significantly fewer tokens. The core contribution is the simplification of memory management into a typed, dense-retrieval framework without complex graph structures or operating-system-style schedulers.

**Reviewer Concerns:**

- The architecture is straightforward, modular, and offers significant latency and token-efficiency gains over full-context methods (Reviewers n2Bc, Mn4m, 38T1).

- The method demonstrates good performance on the selected conversational benchmarks (LoCoMo and LongMemEval).

- The paper is well-written, and the separation of memory types is intuitive. The authors provided a comprehensive rebuttal that clarified the multi-label nature of the router and provided additional ablations regarding the contribution of specific memory stores.

**Reviewer Scores:**

There are significant reservations shared by the reviewers, particularly regarding the scope of evaluation and the mechanical robustness of the memory system.

- The system is effectively "append-only." While the authors argue that typed separation makes future editing easier, the current implementation lacks mechanisms for information updates, conflict resolution, or deletion. Relying solely on timestamps and the LLM's inference capabilities to resolve contradictions between old and new memories is insufficient for a robust "memory orchestration" system. As noted in the post-rebuttal discussion, merely retrieving conflicting chunks and hoping the model sorts them out via timestamps does not constitute a complete solution for long-horizon agentic consistency.

- Multiple reviewers (n2Bc, 38T1) noted that the evaluation is restricted to synthetic conversational QA benchmarks (LoCoMo, LongMemEval). While these are valid for specific sub-tasks, they do not fully substantiate claims regarding general agentic capabilities. Specifically, the utility of "procedural" memory remains under-tested; without evaluation on workflow-oriented benchmarks, it is difficult to validate whether the procedural store actually aids in executing procedures or simply acts as another textual retrieval bucket.

- Reviewer n2Bc pointed out that the specific separation of memory types and the routing strategy seem somewhat heuristic. While the authors provided cognitive science citations in the rebuttal, the technical realization limits the system's ability to handle complex, ambiguous, or evolving user intents compared to learned or feedback-driven approaches.

---

### Decision · Program_Chairs · 2026-01-26

Reject